# Change in the North Atlantic circulation associated to the mid-Pleistocene transition

**Gloria M. Martin-Garcia[1,], Francisco J. Sierro[1], José A. Flores[1], Fátima Abrantes[2]**

[1] Department of Geology, University of Salamanca, Salamanca, Spain

[2] Marine Geology and Georesources (DivGM), Portuguese Institute for the Sea and Atmosphere, Lisbon, and Centre for Marine Sciences at University of Algarve, Portugal

Correspondence to: G. M. Martin-Garcia (gm.martin@usal.es)

**Abstract**

The southwestern Iberian margin is highly sensitive to changes in the distribution of North Atlantic currents, and to the position of oceanic fronts. In this work, the evolution of oceanographic parameters from 812 to 530 ka (MIS20-MIS14) is studied based on the analysis of planktonic foraminifer assemblages from site IODP-U1385 (37º34.285′N, 10º7.562′W; 2585 mbsl). By comparing the obtained results with published records from other North Atlantic sites between 41 and 55 ºN, basin-wide paleoceanographic conditions are reconstructed. Variations of assemblages dwelling in different water masses indicate a major change in the general North Atlantic circulation during MIS16, coinciding with the definite establishment of the 100-ky cyclicity associated to the Mid-Pleistocene Transition. At surface, this change consisted in the re-distribution of water masses, with the subsequent thermal variation, and occurred linked to the northwestward migration of the Arctic Front (AF), and the increase in the North Atlantic Deep Water (NADW) formation respect to previous glacials. During glacials prior to MIS16, the NADW formation was very weak, which drastically slowed down the surface circulation; the AF was at a southerly position and the North Atlantic Current (NAC) diverted southeastwards, developing steep south-north, and east-west, thermal gradients and blocking the arrival of warm water, with associated moisture, to high latitudes. During MIS16, the increase in the

meridional overturning circulation, in combination with the north-westward AF shift, allowed the arrival of the NAC to subpolar latitudes, multiplying the moisture availability for ice-sheets growth, which could have worked as a positive feedback to prolong the glacials towards 100-ky cycles.

**Keywords:** Mid-Pleistocene Transition (MPT); North Atlantic circulation; North Atlantic Current (NAC); Planktonic foraminifers; Iberian margin; IODP-U1385; Glacials.

**1 Introduction**

Climate in the North Atlantic region is characterized by the continuous poleward heat flow carried out by the oceanic circulation. The Gulf Stream and the North Atlantic Current (NAC) transport warm and salty surface water, originated in the tropical region, towards the polar ocean, the northeast Atlantic, and along the western European margin, transferring heat and moisture to the atmosphere during the process (e.g., McCartney and Talley, 1984; Ruddiman and McIntyre, 1984; Schmitz and McCartney, 1993, Rahmstorf, 1994; Chapman and Maslin, 1999). Surface circulation and associated heat flow is pumped by the sinking of surface water in the subpolar region and formation of the North Atlantic Deep-water (NADW). As a matter of fact, the Atlantic Meridional Overturning Circulation (AMOC) is responsible for ~50% of the total poleward heat advection (Sabine et al., 2004; Adkins, 2013).

The NAC forms the transition zone between the cold and productive waters located north of the Arctic Front (AF) (eg., Johannessen et al., 1994), and the warm and oligotrophic waters from the subtropical gyre in the South. Each water mass has distinct physic-chemical characteristics and specific planktonic foraminiferal assemblages (eg., Bé, 1977; Ottens, 1991; Cayre et al., 1999). Various studies have shown that surface water characteristics in the mid-latitude North Atlantic depend on the strength and position of the NAC and associated oceanic fronts (Calvo et al., 2001; Naafs et al., 2010; Voelker et al., 2010). During Pleistocene glacials, the AF migrated southward into mid-latitude North Atlantic (Stein et al., 2009; Villanueva et

al., 2001), cold polar waters expanded to lower latitudes and the NAC did not reach
as far North as during interglacials (e.g., Pflaumann et al., 2003).
After MIS21, a northwestward shift in the position of the AF began (Hernandez-
Almeida et al., 2013), that culminated at the end of MIS16, in a similar location to
today´s (Wright and Flower, 2002). Coinciding with the final stage of this shift, a major
reorganisation of the meridional overturning circulation developed, related to
increased NADW formation that resulted in deeper and southward penetration of this
mass of water (Poirier and Billups, 2014). Both processes could have been related to
the prolongation of glacials that occurred at the end of the mid-Pleistocene transition
(MPT). This was the transitional period during which the Earth´s climate system
underwent a major change, non-linear 100 ky cycles appeared and superimposed
over the more linear, orbital ones of 41 and 23 ky.
Although there is still no agreement over the initiation of the MPT (e.g., Clark et
al., 2006; Maslin and Brierley, 2015), strong 100 ky cycles are recorded since ~650
ka  (Ruddiman et al., 1989; Imbrie et al., 1993; Mudelsee and Schulz, 1997). Related
with the shift in the AF position, warm and salty surface water could reach subpolar
latitudes during glacials, which would have provided the necessary humidity to
prolong the growth of ice sheets, as well as enhanced meridional overturning – both
processes acting as feedback mechanisms partly responsible for the change of the
climate system phasing (Imbrie et al., 1993). The objective of this work is to study the
evolution of glacial circulation in the North Atlantic from MIS20 to MIS14, and explore
its possible relation with the MPT.
Over the last glacial cycle, the Iberian margin recorded both peak displacement
events of the AF and periods of greater influence of subtropical water from the Azores
Current (AzC) (eg., Martrat et al., 2007; Eynaud et al., 2009; Salgueiro et al., 2010).
There is also evidence that polar to tropical planktonic foraminifers assemblages co-
occurred in a latitudinal band around 35º – 40ºN during the Last Glacial Maximum
(McIntyre et al., 1972), which suggests that the limit between both water masses was
situated slightly southwards than it is today (Fiúza et al., 1998; Peliz et al., 2005). Site
IODP-U1385 (37º34´N) lies within this oscillating boundary, and has been shown an
ideal location to study oceanographic changes in the North Atlantic through glacial-
interglacial periods (e.g., Maiorano et al., 2015; Martin-Garcia et al., 2015; Rodríguez-
Tovar et al., 2015; Rodrigues et al., 2017). Analyses of planktonic foraminifer
assemblages are used to identify the different water masses, and results from IODP-
U1385 are compared with published data from other North Atlantic latitudes to reach
basin-wide conclusions.

**2 Materials and Methods**
**2.1 IODP Site U1385**
The Southwestern Iberian margin is a focal location for paleoclimate and
oceanographic research of the Quaternary (Hodell et al. 2013). Site IODP-U1385 was
drilled at the so-called Shackleton Site (37º34.284´N, 10º7.562´W), at 2589 meters
water depth (Fig. 1). At the surface, this area lies under the influence of the *North*
*Atlantic Central Water* (NACW), with a complex circulation pattern; at depth, the
NADW flows between ~2,200 and 4,000 meters, above the *Antarctic Bottom Water*
(AABW).
Today´s surface water circulation in the North Atlantic (Fig. 1a) consists of two
different branches. The NAC, after reaching the subpolar ocean, drifts southwards
along Europe transporting the Eastern North Atlantic Central Water of sub-polar origin
(ENACWsp), formed north of 46º (Brambilla and Talley, 2008). In the south, the AzC,
of subtropical origin (ENACWst) and formed along the Azores Front (Rios et al.,
1992), drifts eastwards and bifurcates when approaching the continental margin. The
ENACWst is saltier, warmer, less dense than the ENACWsp and overflows it along
Iberia with a decreasing lower limit from south to north until  ~42.7 ºN (Fiúza et al.,
116  1998).
Sediments at Site U1385 define a single, very uniform, lithological unit.
Calcareous muds and calcareous clays dominate the lithology. The relative
proportions of carbonate (23% - 39%) and terrigenous materials show in the sediment
color that varies from dark (i.e., more terrigenous) to light (i.e., more calcareous). The
average sedimentation rate for the section is of ~10 cmky$^{-1}$ (Stow et al., 2012).

**2.2 Foraminiferal study**

This study covers a section comprised between 67.2 and 94.6 crmcd (MIS14 - MIS20). The age model (Hodell et al., 2015) is based on the correlation of the benthic oxygen isotope record to the global benthic LR04 isotope stack (Lisiecki and Raymo, 2005). For better comparing our results with data from other North Atlantic sites, new age models were calculated for sites 980 and 607, based on correlations with the LR04 stack.

Sampling was performed every 20 cm, providing a 1.76–ky resolution on average. A total of 147 samples, 1 cm-thick, were freeze-dried, weighed and washed over a 63-$\mu$m mesh. The >63 $\mu$m residue was dried, weighed and sieved again to separate and weigh the >150 $\mu$m fraction. Planktonic foraminifers' taxa were identified (Kennett and Srinivasan, 1983) in aliquots of this last fraction containing a minimum of 300 specimens.

The microfaunal analysis focused on species and assemblages that are associated with North Atlantic surface water masses (Appendices A and B). *Neogloboquadrina pachyderma* sinistral (*N. pachyderma* sin) is an indicator of polar water (Cayre et al., 1999; Pflaumann et al., 2003; Eynaud et al., 2009). *Turborotalita quinqueloba* dwells in cold waters and is usually associated with the AF (Johannessen et al., 1994; Cayre et al., 1999). *Globigerina bulloides, Globigerinella siphonifera (aequilateralis), Globorrotalia inflata*, and *Neogloboquadrina incompta* (former *N. pachyderma* dextral), form the North Atlantic Current (NAC) assemblage, as defined by Ottens (1992). Finally, species included in the warm surface assemblage (Vautravers et al., 2004) are: *Beela digitata, Globigerina falconensis, Globigerinella siphonifera (aequilateralis), Globigerinoides ruber, Globigerinoides sacculifer, Globoturborotalita rubescens, Globoturborotalita tenella, Orbulina universa,* and *Pulleniatina obliquiloculata*.

**2.3. Estimation of thermal gradients**

Thermal gradients in the North Atlantic are reconstructed by calculating the difference between the Sea Surface Temperature (SST) from two sites. The site 607 was used as start point, and compared with sites 980 for the latitudinal gradient ($SST_{607} - SST_{980}$), and U1385 for the longitudinal one ($SST_{607} - SST_{U1385}$). In this

way, a positive longitudinal gradient means that SST was warmer at site 607 than at
U1385; a negative longitudinal gradient indicates warmer SST off SW Iberia than at
site 607.
This estimation of thermal gradients is possible because all the SST records used
for this work are based in planktonic foraminifers´ census counts. Nevertheless,
previous to the comparison, interpolation was applied to obtain records with the same
age points.

## 163 3 Results

Except in the eighth climate cycle (MIS19-MIS18), *Neogloboquadrina*
*pachyderma* sinistral does not vary at glacial-interglacial scale, but peak percentages
are associated either with glacial maxima (MIS20) or to deglaciations, both
Terminations and other deglacial events (Fig. 2b), revealing increased advection of
polar water at these times. *N. pachyderma* sin is less abundant during interglacial
conditions than during glacials, but it is important to note that its percentage during
glacials change through the time series. This species is more abundant during
glacials MIS20, MIS18 (when the highest percentages occurred), and the first half of
MIS16, than during late MIS16 and glacial MIS14 (Fig. 2b). After ~650 ka, *N.*
*pachyderma* sin stays below 10%, except during deglacial events MIS15b/a and at
the end of MIS15, as inferred from sharp decreases in $\delta^{18}$O (Fig.2a-b). This suggests
that since mid-MIS16, the polar water only reached the southwest Iberian margin
associated to some deglacial episodes, and not during full glacial conditions or glacial
maxima, in opposition to what happened before ~650 ka.
*Turborotalita quinqueloba* shows lower percentage during MIS20 and MIS18,
than since MIS16 (Fig. 2c). Highest values occur at ~650 ka and during MIS15b, the
glacial interval that interrupted interglacial MIS15. The variation of *T. quinqueloba* in
site U1385 does not show an interglacial-glacial pattern, which suggests this site did
not register the migration of the AF through each climate cycle.
The NAC assemblage (Ottens, 1992) is the most abundant one at this site (Fig.
2), indicating that the ENACWsp dominates the surface oceanography in the area
through the time series. This assemblage does not keep a similar interglacial-glacial
pattern through the whole study interval, but changes its behaviour at ~650 ka.
Previous to ~650 ka, its variation mirrors that of *N. pachyderma* sin, and the highest
values occur during interglacials. In opposition to this, since ~650 ka, the highest
percentages coincide with full glacial conditions (MIS16a and MIS14a), not with
interglacials (Fig. 2d).
The Warm Surface (WS) assemblage (Vautravers et al., 2004) is typical of the
subtropical water transported eastwards by the AzC. In U1385, this assemblage
shows a clear interglacial-glacial pattern only since Termination TVIII, its percentage
decreasing gradually during MIS17-16 until the glacial maximum (Fig. 2e). Comparing
glacial stages, MIS20 records the highest average relative abundance (16.8%) and
MIS14, the lowest (8.7%). Termination TIX records the most abrupt decrease of this
assemblage (15% drop), while at TVI it even increases (5% rise). At the beginning of
each interglacial, the percentage of this assemblage rises rapidly, suggesting that the
AzC strengthens rapidly in the area after Terminations.

**5 Discussion**
The location of sites 607 and 980 along the main core of the NAC towards the
high latitudes of the North Atlantic (Fig. 1a), allowed us to monitor past changes in the
northward heat transport, using planktonic foraminifer assemblages and SST
reconstructions from both sites. By contrast, planktonic foraminifer assemblages at
site U1385 are more influenced by the advection of heat to the northeastern Atlantic
through the easternmost branches of the NAC, and especially by the AzC, that
originates in the tropics and flows towards Iberia following the northern margin of the
subtropical gyre (Fig. 1a). In consequence, with these three strategic sites, we can
monitor changes in the main circulation systems of the NE Atlantic during the mid-
Pleistocene, and estimate the heat advection to the north (SST gradient between
sites 607 and 980) and to the northeast Atlantic (SST gradient between sites 607 and
U1385) (Fig. 3f-g).

**5.1 North Atlantic circulation during glacials MIS20 and MIS18**

During both glacials, progressive cooling is recorded in sites 607 and 980 (Fig. 3f). Though the cooling is more pronounced at the higher latitude, the SST gradient between both sites is not very high and decreases largely towards the end of glacial stages (Fig. 3g). In contrast, the Iberian margin remained relatively warm during most of MIS20 and a large part of MIS18 (Fig. 3f), which undoubtedly reflects a continuous flow of the AzC to this region, as also indicated by the WS assemblage record (Fig. 2e).

At the subpolar latitude of site 980, the presence of polar water increased rapidly since glacial inceptions, as informed by very high percentages of *N. pachyderma* sin during MIS20, MIS18e, and MIS18a (Fig. 3c). As glacial conditions progressed, the heat flow along the main core of the NAC reduced largely, and even interrupted at glacial maxima MIS20a and MIS18a, as can be inferred from the low temperatures registered in the Azores region (site 607, Fig. 3f). This reduced advection of warm water from the tropics to subpolar latitudes triggered the southward migration of the AF, that surpassed 50 ºN during both MIS20, MIS18e, and MIS18a (Wright and Flower, 2002), and favoured the advection of polar water as far south as site 607, as informed by the record of *N. pachyderma* sin (Fig. 3c).

While the northward flow of heat decreased progressively along both glacials, the heat flow towards the Iberian margin continued in the early part of glacial MIS18 and, especially, during MIS20, indicating a very active AzC during both glacials. This current advected warm water eastward, and deflected northward along the Iberian margin, similarly to today´s IPC (Fig. 1a), probably overflowing the polar water mass, as the co-occurrence of polar and subtropical fauna suggest (Fig. 2b,e). The advection of the warm AzC to site U1385 was only interrupted at Terminations TIX, TVIII, and at deglaciation MIS18e/d, when massive surges of very cold and low-salinity surface waters reached the area, which was registered by peaks of the polar species *N. pachyderma* sin and sharp decreases in the WS assemblage (Fig. 2b,e). This interpretation is corroborated by the negative longitudinal thermal gradient between sites 607 and U1385 (Fig. 3g), which indicates that, an important fraction of the heat reaching the Iberian margin did not flow through the site 607 region.

The very low SST at the mid-latitude site 607, and the low latitudinal thermal
gradient, during glacial maxima MIS20a, MIS18e and MIS18a (Fig. 3f-g), suggests
either a complete shut-down of the NAC core flux, or a southward or southeastward
diversion of this current,.as glacial conditions progressed. Nevertheless, the low
thermal gradient between sites 607 and U1385 (Fig. 3g) implies that the SW Iberian
margin was always under the influence of the warmer AzC.

**5.2 Changes in the North Atlantic circulation starting at MIS17**
Both latitudinal and longitudinal thermal gradients (Fig. 3g) inform of drastic
rearrangement of North Atlantic circulation starting at MIS17. SST at site 607 was
much warmer than during MIS19, although both interglacials were similar, according
to $\delta^{18}O$ (Fig. 3a,f). This points to a reactivation of the NAC during MIS17, and a
displacement of this current westward site 607. Such reactivation would be the result
of increased NADW formation, that reached higher rates than during the previous
interglacial, as suggested by the ~0.2‰ higher $\delta^{13}C$ in MIS17 than in MIS19 (Fig. 3b).
On the other hand, the very high latitudinal thermal gradient (Fig. 3g) suggests that
this current did not reach subpolar latitudes, as it did during the following interglacial,
MIS15, when this gradient was much lower.
The unusually high longitudinal thermal gradient registered during MIS17 was due
to the prolonged deglaciation of MIS18, that continuously advected polar water along
the Iberian margin (Martin-Garcia et al., 2015), resulting in very cold SST and high
percentages of *N. pachyderma* sin, at site U1385 (Fig. 3).
MIS16 was a very prolonged glacial with extensive ice sheets; nevertheless, polar
waters did not extend to the mid-latitude ocean, as suggested by the low percentages
of *N. pachyderma* sin in sites 607 and U1385 (Fig. 3c).
The latitudinal thermal gradient for most of MIS16, and the whole MIS14, was
notably higher than during MIS20-18 (Fig. 3g). This great SST decrease, between
sites 607 and 980, must be the result of a significant heat loss to the atmosphere and
associated release of water vapour, along the path of the NAC during both MIS16 and
MIS14. This water vapour release provided the necessary moist to continue ice-
sheets growth, opposite to what had happened during previous glacials. Also contrary

to glacials MIS20 and MIS18, when the surface water at the subpolar site 980 progressively cooled towards glacial maxima without important millennial-scale oscillations (Fig. 3f), in glacials MIS16 and MIS14, the surface ocean circulation was very variable and the AF migrated northward-southward site 980 very frequently (Fig. 3c-d). During short time periods, the NAC reached this subpolar site, conveying heat to the northern-latitude Atlantic (Fig. 3e). However, this oscillation of the AF never affected middle latitudes, according to the fairly mild SST, and low percentage of *N. pachyderma* sin, recorded both in the open ocean and in the continental margin during MIS16-14 (Fig. 3c,f).

In the mid-latitude ocean site 607, SST during MIS16 and MIS14 were very different from those recorded in MIS20 and MIS18 (Fig. 3f). While in the older glacials SST decreased towards glacial maxima, this trend is not observed during MIS16 and MIS14, and warm SST was recorded also during glacial maxima.

Although warmer SST were recorded through the mid-latitude North Atlantic, a negative thermal gradient still prevailed during MIS16-14, between sites 607 and U1385 (Fig. 3g), indicating a continuous heat flow toward southwest Iberia. This suggests that, this region remained under the influence of the subtropical AzC during most part of glacials MIS16 and MIS14, as it also did during MIS20, based on the mild SST registered at that time (Fig. 3f). Contrary to previous glacials, the NAC kept vigorous in site U1385 during MIS16, except at ~655 ka, and MIS14, and increased its strength as glacials advanced (Fig. 2d).

**5.3 Implications of changes in the North Atlantic circulation associated with the MPT**

Assuming a close correlation between the rate of AMOC and benthic $\delta^{13}$C levels (Zahn et al, 1997; Adkins et al., 2005; Hoogakker et al., 2006), we interpret that the published $\delta^{13}$C data from the sub-polar North Atlantic (Wright and Flower, 2002; Hodell et al., 2008; Hodell and Channell, 2016) document a long-term increase in the NADW formation rate, that initiated in MIS22 and culminated in MIS14. Since MIS17, mid-latitude and subtropical North Atlantic sites registered a progressive increase of

NADW at depths previously occupied by the AABW ($\delta^{13}$C data in e.g., Poirier and
Billups, 2014; Martin-Garcia et al., 2015).
The increased production of NADW, during glacials after MIS16 respect to
previous ones, triggered the advection of relatively-warm NAC towards subpolar
latitude, providing additional humidity to the area and, thus, enhancing the growth of
ice sheets, which led to the prolonged and extreme glaciation of MIS16, one of the
first and most prominent glacials of the "100-ky world". In addition, the intermittent
advection of this warm water made ice sheets more vulnerable to internal instabilities,
with the subsequent release of icebergs registered in the North Atlantic during MIS16
(e.g., Wright and Flower, 2002; Hodell et al., 2008). The interaction between a more
intense AMOC and ice sheet instabilities, registered by rapid migrations of the AF
north and south of site 980 (Fig. 3c-d), resulted in punctual events of sharp reduction
of the NADW formation, like that at ~655 ka that coincided with one of the
southernmost positions of the AF, according to the record of *T. quinqueloba* in site
980 (Wright and Flower, 2002), and was also registered in U1385 by peaks in this
species and in *N. pachyderma* sin, coinciding with very low percentage of NACass
(Fig. 3b-e). Both this episode and the outstanding one ~650 ka, with the lowest $\delta^{13}$C
value since MIS18 in middle latitudes in coincidence with very high abundance of the
NACass in high latitudes (Fig. 3b,e), points to an exceptionally vigorous but shallow
NA overturning cell, underlain by significant volumes of southern-sourced water,
similarly to the situation at the end of TII (Böhm et al., 2014). This mode of AMOC,
according to benthic $\delta^{13}$C records, maintained during glacial stages MIS16, MIS15b,
and MIS14, when the subpolar site 980 recorded > 0.25 ‰ higher $\delta^{13}$C than southern-
more sites (Wright and Flower, 2002; Martin-Garcia et al., 2015; Hodell et al., 2016).
This vigorous AMOC mode recorded in MIS14 was the culmination of a sequence
of increasing deepening of the overturning circulation cell that initiated in MIS22, and
was registered by a tendency towards higher benthic $\delta^{13}$C, both in high and mid-
latitude sites U1308 and U1313, from MIS22 to MIS14 (Hodell and Channell, 2016),
and was especially noticeable during glacial stages. During MIS20 and MIS18, ice
sheets collapses (Wright and Flower, 2002) produced a continuous flux of meltwater
pulses that kept very weak NADW formation; the deep North Atlantic being occupied
by southern-sourced waters, according to very low benthic $\delta^{13}C$ recorded both in
middle and high latitudes (Wright and Flower, 2002; Hodell et al., 2015; 2016). During
these glacials, the almost shutdown AMOC maintained the AF at a southern position
and prevented the northward flux of the necessary moisture for the growth of ice
sheets, which could not work as a positive feedback and extend glacial stages over
obliquity and precessional (41- and 23 ky) cycles, as they worked during MIS16, one
of the first and most prominent glacials of the "100-ky world".

**6 Conclusions**
By studying planktonic foraminiferal assemblages from the Iberian margin (IODP-
U1385) for the interval 812–530 ka and comparing them with records from other sites
between 41 and 55 ºN, we are able to trace paleoceanographic conditions across the
North Atlantic from MIS20 to MIS14 and draw the following conclusions:
Variations of microfaunal assemblages associated to surface currents indicate a
major change in the general North Atlantic circulation during this interval, coinciding
with the definite establishment of the 100-ky climate phasing. In surface, this change
consisted in the re-distribution of water masses and associated SST that happened
linked to the northwestward migration of the AF during MIS16, and was related with
the increasing NADW formation trend that initiated in MIS22.
Prior to MIS 16, the AMOC rate was very low, especially during glacials, the AF
was at a southerly position, and the NAC diverted southeastwards, developing steep
south-north and east-west thermal gradients, and blockading the arrival of warm
water, with associated moisture, to the high latitude North Atlantic.
During MIS16, the NADW formation increased respect to previous glacials,
especially during glacial maxima, which resulted in the north-westward AF shift and
enhanced surface circulation, allowing the arrival of the relatively-warm NAC to
subpolar latitudes and increasing the moisture availability to continuing the ice sheets
growth, which would have worked as a positive feedback to prolong the duration of
glacials to 100-ky cycles.


**Appendix A:** Planktonic foraminifer species used in this study

| Species | Environment | References |
|---|---|---|
| *Neogloboquadrina pachyderma* sinistral (Ehrenberg 1861) | Polar | Pflaumann et al. (1996); Cayre et al. (1999); Schiebel and Hemleben (2017) |
| *Turborotalita quinqueloba* (Natland 1938) | Subpolar | Ottens (1991); Schiebel and Hemleben (2017) |
| *Globigerina bulloides* d´Orbighy 1826 | NA current<br>Transitional | Ottens (1991)<br>Schiebel and Hemleben (2017) |
| *Neogloboquadrina incompta* (Cifelli 1961) (Previously known as *N. pachyderma* dextral) | NA current<br>Portugal current | Ottens (1991)<br>Salgueiro et al. (2008) |
| *Globorrotalia inflata* (d´Orbigny 1839) | NA current<br>Portugal current<br>Transitional | Ottens (1991)<br>Salgueiro et al. (2008)<br>Schiebel and Hemleben (2017) |
| *Globigerinella siphonifera* (d´Orbighy 1839) | Azores current<br>Warm surface<br>Subtropical | Ottens (1991)<br>Vautravers et al. (2004)<br>Schiebel and Hemleben (2017) |
| *Beela digitata* (Brady 1879) | Warm surface<br>Subtropical | Vautravers et al. (2004)<br>Schiebel and Hemleben (2017) |
| *Globigerina falconensis* Blow 1959 | Warm surface<br>Subtropical | Vautravers et al. (2004)<br>Schiebel and Hemleben (2017) |
| *Globigerinoides ruber* (d´Orbighy 1839) | Subtropical<br>Warm surface<br>Azores current<br>Subtropical / tropical | Ottens (1991)<br>Vautravers et al. (2004)<br>Salgueiro et al. (2008)<br>Schiebel and Hemleben (2017) |
| *Globigerinoides sacculifer* (Brady 1877) | NA transitional<br>Warm surface<br>Tropical | Ottens (1991)<br>Vautravers et al. (2004)<br>Schiebel and Hemleben (2017) |
| *Globoturborotalita rubescens* Hofker 1956 | Azores current<br>Warm surface<br>Subtropical | Ottens (1991)<br>Vautravers et al. (2004)<br>Schiebel and Hemleben (2017) |
| *Globoturborotalita tenella* (Parker 1958) | Azores current<br>Warm surface<br>Subtropical | Ottens (1991)<br>Vautravers et al. (2004)<br>Schiebel and Hemleben (2017) |

| *Orbulina universa* d´Orbigny 1839 | Warm surface | Vautravers et al. (2004) |
|---|---|---|
| | Subtropical | Schiebel and Hemleben (2017) |
| *Pulleniatina obliquiloculata* (Parker and Jones 1865) | Azores current | Ottens (1991) |
| | Warm surface | Vautravers et al. (2004) |
| | Tropical | Schiebel and Hemleben (2017) |

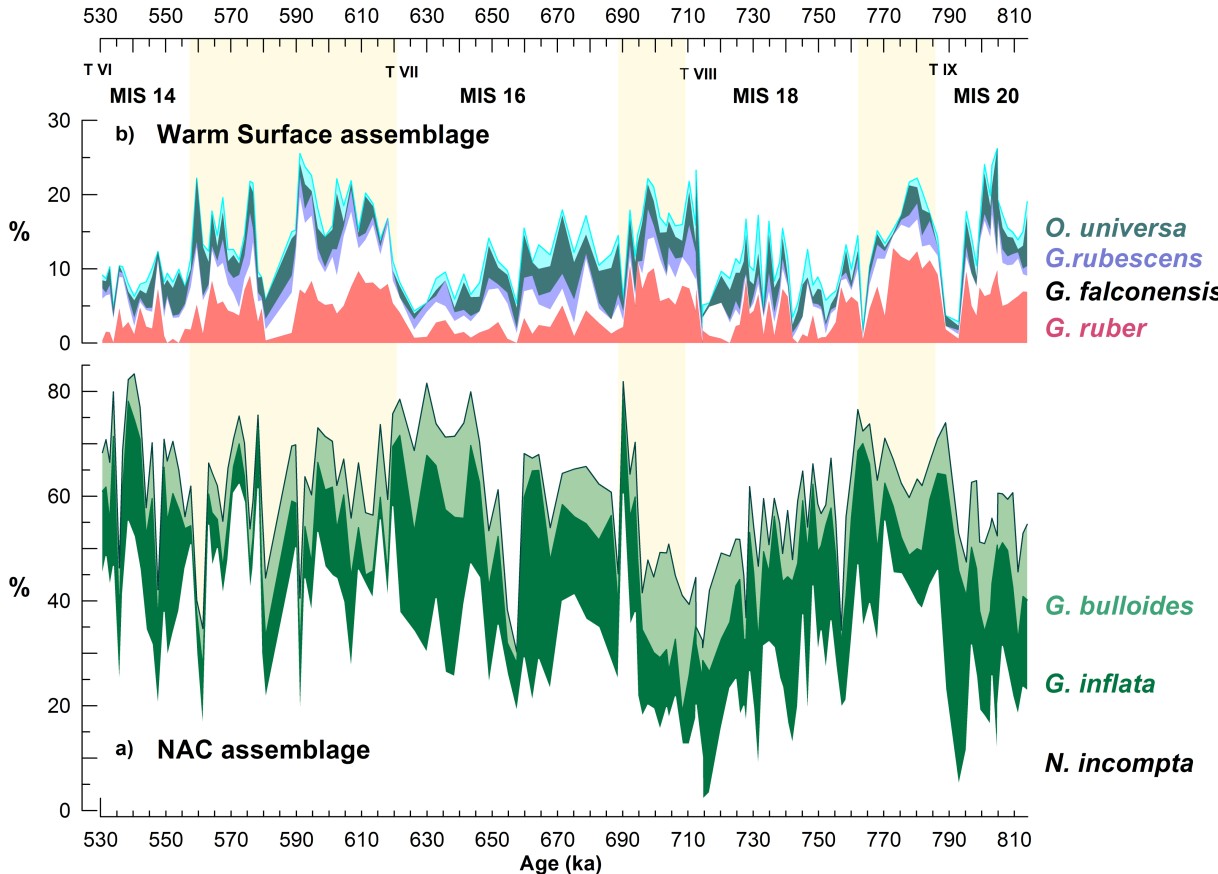

**Appendix B:** Faunal composition of both the NAC, and the warm surface assemblages in site U1385 through the study interval. (a) *N. incompta* (white), *G. inflata* (dark green) and *G. bulloides* (light green). (b) *G. ruber* (red), *G. falconensis* (white), *G. rubescens* (lilac), *O. universa* (dark green), and in cyan, other species with less than 1.5% each: *G. siphonifera*, *G. tenella*, *B. digitata*, *G. sacculifer* and *P. obliquiloculata*.

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

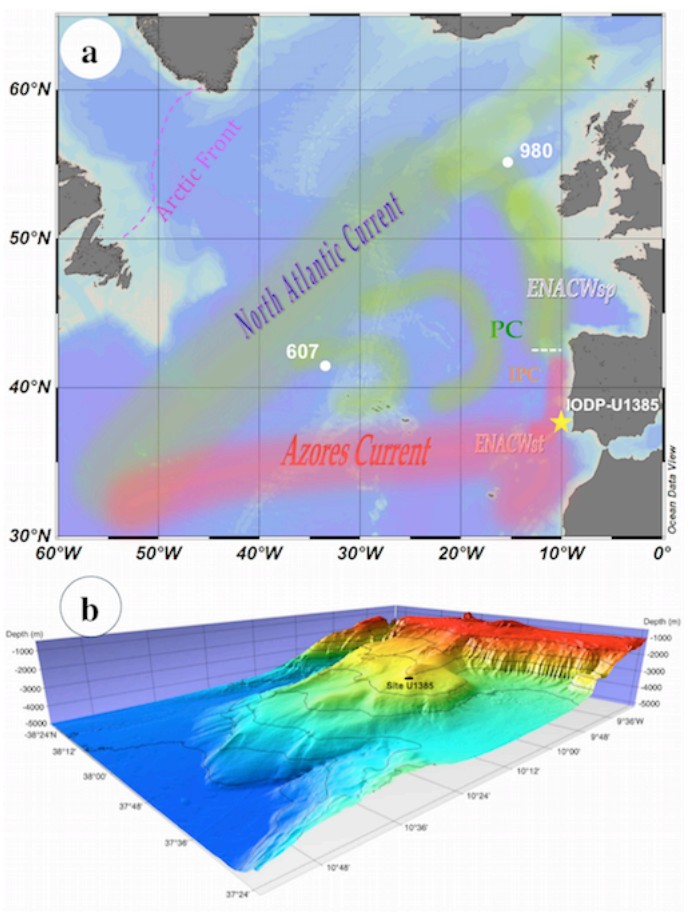

**Figure 1.** (a) Modern surface circulation in the North Atlantic and location of IODP-
U1385 and other sites discussed in this paper. *ENACWsp* Eastern North Atlantic
Central Waters of subpolar origin; *ENACWst*, Eastern North Atlantic Central Waters
of subtropical origin; *IPC*, Iberian Poleward Current; *PC,* Portugal Current. The white
dashed line represents the today´s approximate surface limit between *ENACWsp* and
*ENACWst* (Fiúza et al., 1998). (b) Regional bathymetry of the SW Iberian margin,
showing site U1385 (Expedition 339 Scientists, 2012).

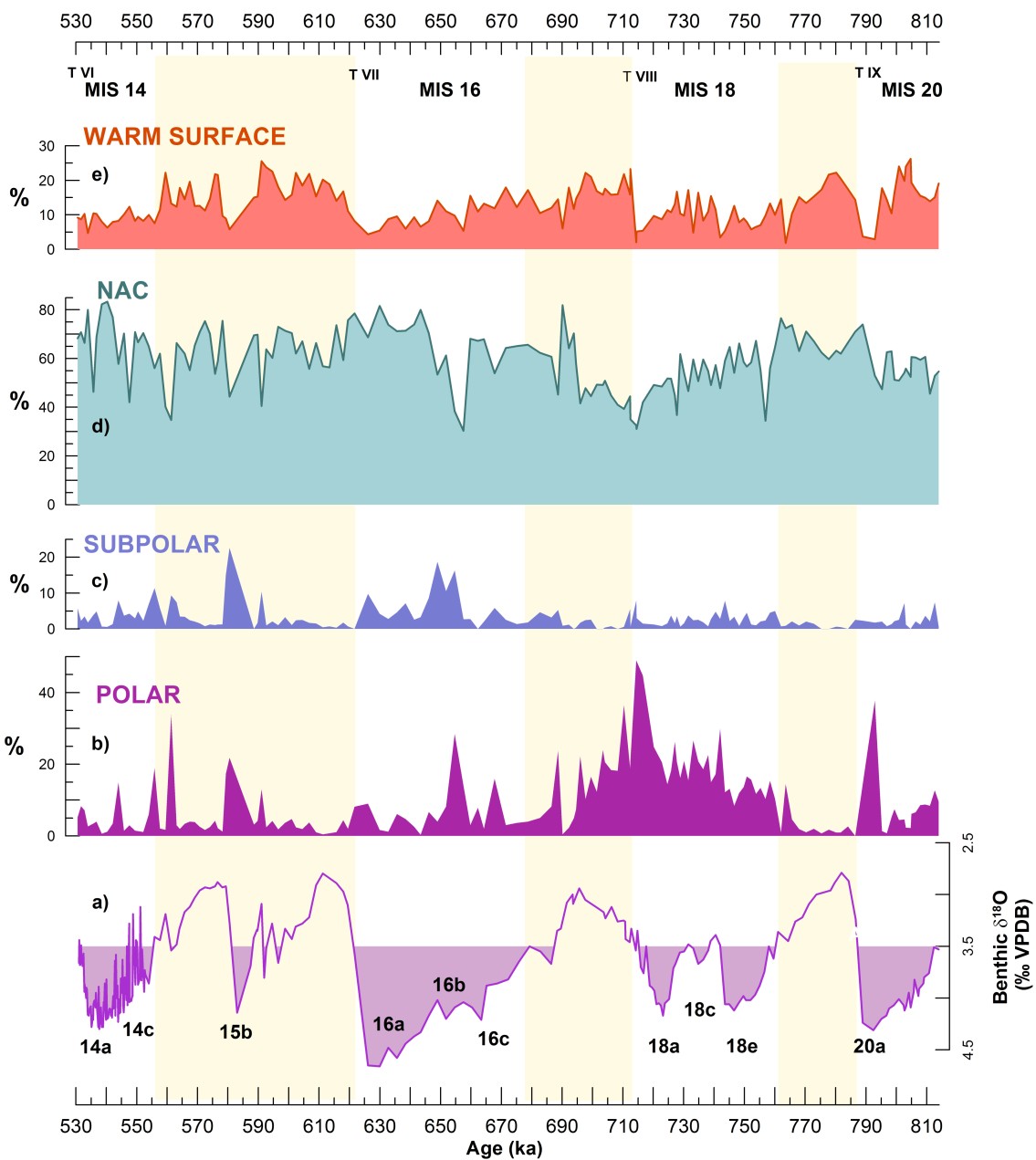

**Figure 2.** Relative abundance of planktonic foraminiferal species and assemblages in IODP-U1385 through MIS 14-20, and comparison with benthic isotope data from the same site. (a) Benthic $\delta^{18}O$ record (Hodell et al., 2015) with filling enhancing glacial conditions according to the threshold for the North Atlantic (McManus et al., 1999); glacial substages are named according to Railsback et al. (2015). Relative

abundance of: (b) polar species *N. pachyderma* sinistral; (c) subpolar species *T. quinqueloba*; (d) NAC assemblage (as defined by Ottens, 1991); and (e) warm surface assemblage (as defined by Vautravers et al., 2004). Yellow bands highlight interglacials. Terminations (T) are marked in roman numerals. IODP-U1385 isotopic record is from Hodell et al. (2015).

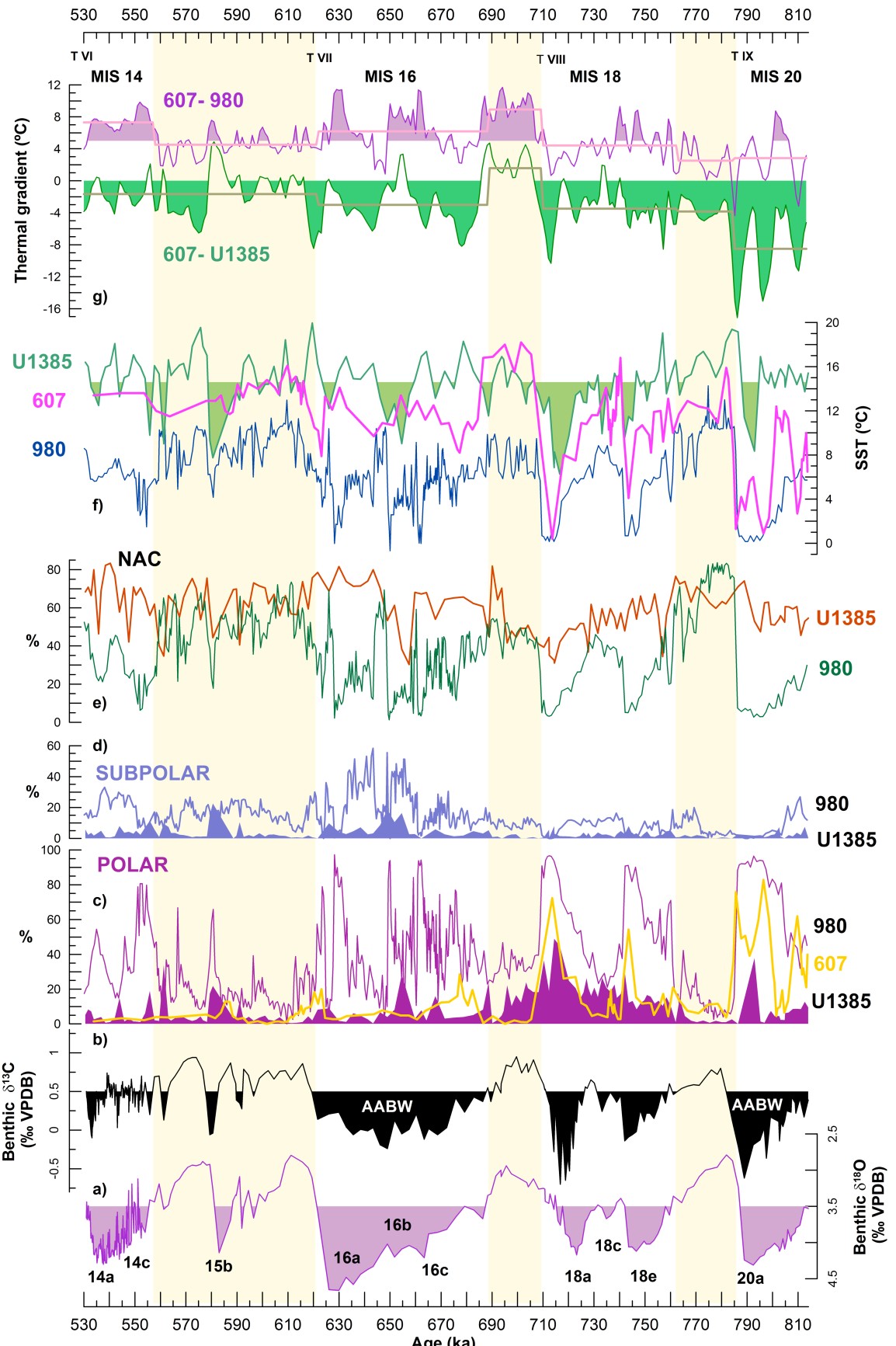

580

**Figure 3.** Comparison of records from the mid-latitude (IODP-U1385; ODP-607) and the subpolar (ODP-980) North Atlantic. Benthic $\delta^{18}O$ (a), and $\delta^{13}C$ (b) from U1385 (Hodell et al., 2015); filling in (b) enhancing $^{13}C$-depleted values typical for Antarctic bottom water (AABW) (Adkins et al., 2005). (c) Percentage of *N. pachyderma* sinistral in sites U1385 (filled), 607 (glod) and 980 (purple). (d) Relative abundance of *T. quinqueloba* for sites U1385 (filled) and 980. (e) Relative abundance of the NAC assemblage (as defined by Ottens, 1991) in sites U1385 (red) and 980 (green). Site 980 faunal data are from Wright and Flower, 2002; for this work, the NAC assemblage of site 980 has been calculated using the published census counts. (f) SST from sites 980 (dark blue; Wright and Flower, 2002), 607 (pink; Ruddiman et al., 1989), and U1385 (green; Martin-Garcia et al., 2015), with filling enhancing lower than 14.6 ºC, the average SST for the study interval. (g) Longitudinal (green) and latitudinal (purple) thermal gradients, with the statistical mean for each MIS represented in superimposed straight lines. Age models for sites 980 and 607 have been re-calculated using the LR04-stock. Yellow bands highlight interglacials. Terminations (T) are marked in roman numerals.