# Peer review of "Gloria M. Martin-Garcia1,, Francisco J. Sierro1, José A. Flores1, Fátima Abrantes2"

_Climate of the Past, 2018_

## Referee Comment (RC1) · Anonymous Referee #1 · 13 Apr 2018

General Comments Martin-Garcia use foraminiferal assemblages from a drill core in the eastern subtropical Atlantic (U1385) to reconstruct glacial/interglacial variations in surface hydrography during the latter stages of the mid-Pleistocene transition (MPT) (Marine Isotope Stages 20-14, ∼800-530Ka). Their overall conclusion, as stated in the abstract, is that beginning with MIS 16, there is an increase in poleward warm surface water transport in the North Atlantic Current, which would act as a positive feedback prolonging the more extreme, 100 kyr-paced glacial maxima.

The MPT is at the center of much research because the underlying mechanisms responsible for the evolution of the 100 kyr cycle are not well understood, although a number of hypothesis exist. Therefore, the manuscript is of timely importance contributing new information about the state of the surface ocean during this interval of

time.

The current version of the manuscript text is not written in a way that makes it easy to evaluate whether or not the data support the major findings. The Results and Discussion sections need reorganization to better highlight how the data lead to the stated conclusions. I suggest describing all time series to guide the reader through the study. In the context of Figure 3, why not show the N. pachy counts from Site 607? Interpretations regarding heat transport are based on spatial thermal gradients, yet none of the figures show such gradients. The reader is asked to figure this out from the SST records shown in Figure 4. It is also really difficult to follow the argumentation in the discussion because statements are not followed-up with appropriate call-outs to figures.

There are a few statements in the text that seem to go against what is generally known about deep water circulation on glacial/interglacial time scales. For example, in the abstract the authors imply that NADW is strong during MIS 16 (lines 20-27)? To my knowledge, and shown in Figure 3b, the relative flux of NADW increased during the deglaciation. So perhaps this is just a matter of carefully rewording the pertaining sentences. There are numerous other instances in the text where the wording of the sentences does not clearly communicate the message (see details below).

Interpreting changes in percentages is complicated by the fact that an increase in one species results in an apparent decrease in another, when, in fact, there may not be a change at all in the accumulation of the latter species. The authors should address this so-called 'closed sum' problem.

Regarding the description of deep water mass changes, I suggest rewording the sentences to make it clear that it is the relative fluxes of NADW and AABW that are changing.

How do these results compare with Alonso-Garcia et al. (2011) specifically? The time intervals of study overlap, so there is potential to make more of this comparison. Or,

are the interpretations of the shifting fronts based on their findings? In this case the study should be cited in the discussion.

Specific Comments

Lines 59-62: include Alonso-Garcia et al 2011 in the list of citations?

Line 64: Alonso-Garcia's record begins with MIS 19. Therefore, it is no entirely appropriate to cite their study in the context of something that "began" during MIS 21?

Line 66: Why abbreviate the reference to Wright and Flower (2002) with W&F02? None of the other citations are abbreviated.

Line 92: "to obtain an conclusion" seems awkward. Perhaps replace with: to reach basin-wide conclusions? Or to obtain a basin-wide picture/view/reconstruction?

Line 96-97: Awkward sentence. Do you mean that the records extend far back into the past, or that they have been studied for a long time?

Line 134: "generally present" is vague. Figure 2b shows that N. pachy are present throughout the entire study interval, but their relative abundance increases during these glacial intervals. I suggest specifying what percentages are considered significant and why. For example, there is also a peak during MIS 15.

Line 149-151: This reads as if you are implying that MIS 20 is an interglacial interval.

Line 156: I would suggest changing the section heading to specify that the focus is on MIS 20 and MIS 18

Line 169-170: The sentence needs a specific figure call-out. I found the info in Figure 4c and d.

Line 181: Vague: What is the difference between very low and relatively low? And, it is confusing to read about low ice volume in the context of glacial intervals (MIS 20 and 18).

Line 192: Define what the thermal gradient is. What does it mean when it is negative in terms of the temperature difference between the sites? Once this is established, it is easier to follow the interpretation with respect to heat transport.

Line 214: I am not sure that I see that the thermal gradient was significantly differ-ent during MIS 18 from MIS 16. This is true only for some intervals of time, but not consistently. For example, the same SSTs are recorded by the sites during MIS 16 at ∼640-650 Ka. In any case, significance, which is a statistical term, is not demonstrated in this data set.

Line 220: It is really difficult to follow how these records show a negative thermal gradient. Would it be possible to just calculate the SST difference between the records to support this point?

Line 226: I am not sure I detect a repeating pattern in the data set. MIS 14 has quite a bit of variability, as you point out, so which pattern are you referring to?

Line 251: Is this correct? Do these studies really show that AABW is reduced during glacial intervals? There is a study by Lang et al., 2016 (Nature Geosciences) that shows % NADW for the past 3 million years. They show that NADW goes to zero, or almost zero during glacial intervals beginning around 0.9 Ma.

Technical Comments

The following is an incomplete list of editorial-type fixes.

Line 23: "At" the surface

Line 30: Blocking

Line 68. . .during interglacial periods

Line 69: related "to"

Line 86: "...which makes it an ideal location. . .."
Line 99 meters. . .. At the surface. . .; at depth. . ..

Line 122: on average

Line 123: commas before and after 1 cm thick?

Line 131: associated with

Line 141: replace 'since then' with 'after'

Line 215: higher

---

## Referee Comment (RC2) · Anonymous Referee #2 · 17 Apr 2018

Dear Natascha Töpfer Copernicus Publications Editorial Support

I hereby you receive my report on the MS "Role of the North Atlantic circulation in the mid-Pleistocene transition" by Martin-Garcia et al. The authors analysed the planktonic foraminiferal patterns of southwestern Iberian margin site IODP-U1385 comparing these data with two key sites of Atlantic ocean as ODP Sites 607 and 980. This study documented important changes in planktonic foraminiferal assemblage during glacial/interglacial oscillation between 812 to 530 ka (MIS20-MIS14). This reconstruction suggests a major change in general North Atlantic circulation during MIS16. In surface, this change consisted in the re-distribution of water masses, with the subsequent thermal variation, and occurred linked to the northwestward migration of the Arctic Front (AF) and the increase in the North Atlantic Deep Water (NADW) formation.

The present version of the manuscript is confused and it is very hard to follow the text with these figures. Systematically, the reader has to jump from one figure to another, when it could be possible to plot the data in one single figure. The authors suggest a possible link of the observed changes with change in ciclicity over the Mid-Pleistocene Transition, but a detail discuss on time-series is missing. Moreover, the study interval corresponds to the end of the Mid-Pleistocene Transition and without a detailed spectral and wevelt analysis on proxy records is very hard to propose in the manuscript a connection with this important, but not well understood, climate transition. In my opinion, the authors have to describe, using a statistical approach on proxy data, difference, similitude and trend between the three sites. This statistical approach could be used also to evaluate possible thermal gradients. The authors plotted as proxies the NAC and WARM SURFACE groups, but the connection with glacial/interglacial cycles is not clear. This is mainly evident for the NAC signal. This signal is characterised by noise and if we exclude, the increase in abundance at ca. 655ka upwards, the signal does not show a particular pattern. The pattern of WARM SURFACE shows a clear strong increase in abundance in correspondence to the onset of interglacial interval. This pattern is not strongly described in the manuscript. There is an explanation for the strong peaks in abundance of N. pachyderma in coincidence of Termination VIII? This peak is in full deglaciation phase. I would like to suggest to add in the methods a description concerning the construction of the planktonic foraminiferal groups used in the manuscript. I am very surprise to see that Globigerina falconensis is considered as part of warm surface assemblage. This species is generally considered as cool water taxon.

In my opinion the strong difference in time resolution of the sites render very difficult the comparison between the T. quinqeloba and N. pachyderma. In addition, where is the distribution of these taxa for site 607? In addition, the strong difference in NAC patterns from site U1385 and site 980 is not well described and in my opinion not discussed in detail. Why Nps is abbreviate? Please write N. pachyderms left coiled – See line 192 Line 168 – the authors reported Fig.4c-e, But where is Figure 3? Line 213 – Are you

sure that the correct figure is 2? I think that the figure to call up is the Fig. 3

My overall conclusion is that the manuscript is suitable for the journal but unfortunately, it needs moderate to major revision.

---

## Author Comment (AC1) · 25 Jul 2018

Referee #1: doi.org/10.5194/cp-2018-30-RC1

The current version of the manuscript text is not written in a way that makes it easy to evaluate whether or not the data support the major findings. The Results and Discussion sections need reorganization to better highlight how the data lead to the stated conclusions. I suggest describing all time series to guide the reader through the study. The manuscript has been changed in the way Referee #1 suggests. The Results and Discussion sections have been modified to better explain our findings and our conclusions. Time series are better described. The present manuscript describes events occurring during interglacials, and not only during glacials, as the first version

did.

In the context of Figure 3, why not show the N. pachy counts from Site 607? Interpretations regarding heat transport are based on spatial thermal gradients, yet none of the figures show such gradients. The reader is asked to figure this out from the SST records shown in Figure 4. It is also really difficult to follow the argumentation in the discussion because statements are not followed-up with appropriate call-outs to figures. The new Figure 3 includes the N. pachyderma sin record from Site 607 (see Fig. 3c, yellow graphic). In this way, comparison with sites 980 and U1385 is clearer. Both latitudinal and longitudinal thermal gradients have been calculated for the North Atlantic, using data from the studied sites. The estimation of such gradients is described in the Methods section, and the gradients themselves are included in Figure 3g. To better highlight the thermal variation along the time series, the statistical mean has been calculated for each MIS, in both latitudinal and longitudinal gradients, and represented in the same figure. Call-outs to figures have been corrected in the text.

There are a few statements in the text that seem to go against what is generally known about deep water circulation on glacial/interglacial time scales. For example, in the abstract the authors imply that NADW is strong during MIS 16 (lines 20-27)? To my knowledge, and shown in Figure 3b, the relative flux of NADW increased during the deglaciation. So perhaps this is just a matter of carefully rewording the pertaining sentences. There are numerous other instances in the text where the wording of the sentences does not clearly communicate the message (see details below). Following suggestion, the text has been changed as follows: "...and the increase in the North Atlantic Deep Water (NADW) formation respect to previous glacials"

Interpreting changes in percentages is complicated by the fact that an increase in one species results in an apparent decrease in another, when, in fact, there may not be a change at all in the accumulation of the latter species. The authors should address this so-called 'closed sum' problem. It is clear that the closed sum effect exists, but there is no better way to show the results about the planktonic foraminifer assemblages. Sev-

eral authors (e.g., Bé, 1977; Ottens, 1991;) have studied present-day North Atlantic water masses and identified the dominant planktonic foraminiferal species (in percentages) for each of them. In the same way, fossil assemblages have been associated to specific water masses (e.g., Cayre et al., 1999; Vautravers et al., 2004; Salgueiro et al., 2008)

SST reconstructions are also based on assemblage's composition (measured in percentages)

Regarding the description of deep-water mass changes, I suggest rewording the sentences to make it clear that it is the relative fluxes of NADW and AABW that are changing. The text has been changed as follows: ". . .mid-latitude North Atlantic sites registered a relative decrease of the AABW during glacials, and subtropical sites recorded the presence of NADW at depths previously occupied by the AABW"

How do these results compare with Alonso-Garcia et al. (2011) specïfically? The time intervals of study overlap, so there is potential to make more of this comparison. Or, are the interpretations of the shifting fronts based on their findings? In this case the study should be cited in the discussion.

Both Alonso-Garcia et al (2011), and Hernandez-Almeida et al (2013) studied site U1314, situated too north-westward for being useful in the study of variations of the NAC through glacials. This site, as well as others located northward 980 - like 984, studied by Wright and Flower (2002) together with the 980, register advances of the AF very early in glacials, both before and after the MPT. Particularly, site U1314 was compared to U1385 in Martin-Garcia et al. (2015), and SST differences between both sites, studied for the interval 780-490 ka. This study demonstrated that the NAC did not reach site U1314 since glacial inceptions, both before and after MIS16. Site 980, on the other hand, lies in the path of the NAC and thus, at a key location to register both advances of the AF and presence of the NAC during glacials, as can be observed in Fig. 3.

Specific Comments

Lines 59-62: include Alonso-Garcia et al 2011 in the list of citations? These lines refer to the mid-latitude NAtlantic, not to the subpolar one, which is why this citation has not been included

Line 64: Alonso-Garcia's record begins with MIS 19. Therefore, it is no entirely appropriate to cite their study in the context of something that "began" during MIS 21? This citation has been removed

Line 66: Why abbreviate the reference to Wright and Flower (2002) withW&F02? None of the other citations are abbreviated. The text has been changed as suggested

Line 92: "to obtain an conclusion" seems awkward. Perhaps replace with: to reach basin-wide conclusions? Or to obtain a basin-wide picture/view/reconstruction? The text has been changed as suggested: "reach basin-wide conclusions"

Line 96-97: Awkward sentence. Do you mean that the records extend far back into the past, or that they have been studied for a long time? The text has been changed: "for paleoclimate and oceanographic research on the Quaternary"

Line 134: "generally present" is vague. Figure 2b shows that N. pachy are present throughout the entire study interval, but their relative abundance increases during these glacial intervals. I suggest specifying what percentages are considered significant and why. For example, there is also a peak during MIS 15. The new Results section explains the variation of this species through the time series, comparing its relative abundances during glacials/interglacials, and also the occurrence of peak percentages

Line 149-151: This reads as if you are implying that MIS 20 is an interglacial interval. The text has been changed: ". . .even more abundant than during interglacials, like in MIS20, when it reaches the highest percentages of the whole study interval".

Line 156: I would suggest changing the section heading to specify that the focus is on MIS 20 and MIS 18 The new heading is: "5.1 North Atlantic circulation during glacials

MIS20 and MIS18"

Line 169-170: The sentence needs a speciïñÁc ïñÁgure call-out. I found the info in Figure 4c and d. The figure call-out has been added. The information is now in the new Fig. 3f

Line 181: Vague: What is the difference between very low and relatively low? And, it is confusing to read about low ice volume in the context of glacial intervals (MIS 20 and 18). This sentence has been removed.

Line 192: DeïñÁne what the thermal gradient is. What does it mean when it is negative in terms of the temperature difference between the sites? Once this is established, it is easier to follow the interpretation with respect to heat transport. The method to calculate the thermal gradient is now fully explained in the "Materials and Methods" section ("2.3. Estimation of thermal gradients"). This section also explains the meaning of a positive and a negative gradient between sites.

Line 214: I am not sure that I see that the thermal gradient was signiïñÁcantly different during MIS 18 from MIS 16. This is true only for some intervals of time, but not consistently. For example, the same SSTs are recorded by the sites during MIS 16 at ∼640-650 Ka. In any case, signiïñÁcance, which is a statistical term, is not demonstrated in this data set. The ambiguous term has been changed. The new Fig. 3g, includes thermal gradients. As the average value has been calculated separately for each stage, it is easier to see that the latitudinal thermal gradient in the NAtlantic was higher during MIS16, and MIS14, than during the whole interval MIS20-MIS18.

Line 220: It is really difïñÁcult to follow how these records show a negative thermal gradient. Would it be possible to just calculate the SST difference between the records to support this point? Thermal gradients have been calculated between the records, and represented in figure 3g

Line 226: I am not sure I detect a repeating pattern in the data set. MIS 14 has quite a bit of variability, as you point out, so which pattern are you referring to? The text has been modified: "While in the older glacials SST decreased towards glacial maxima, this trend is not observed during MIS16 and MIS14, and warm SST was recorded also during glacial maxima".

Line 251: Is this correct? Do these studies really show that AABW is reduced during glacial intervals? There is a study by Lang et al., 2016 (Nature Geosciences) that shows % NADW for the past 3 million years. They show that NADW goes to zero, or almost zero during glacial intervals beginning around 0.9 Ma. We are comparing conditions during glacials. It is proved that there is and increasing trend in the NADW formation rate since MIS22, but it is during glacials that, the difference in the AMOC rate influences the mass of water present in the deep mid-latitude North Atlantic. The text has been changed to explain this better: ". . .data from the sub-polar North Atlantic (Wright and Flower, 2002; Hodell et al., 2008) document a long-term increase in the NADW formation rate, that initiated in MIS22 and culminated in MIS14. This enhanced the southward flux of the NADW and, since MIS17, mid-latitude North Atlantic sites registered a relative decrease of the AABW during glacials, and subtropical sites recorded the presence of NADW at depths previously occupied by the AABW (e.g., Poirier and Billups, 2014; Hodell et al., 2015)".

Technical Comments The following is an incomplete list of editorial-type fixes. Line 23: "At" the surface Line 30: Blocking Line 68: during interglacial periods Line 69: related "to" Line 86: "...which makes it an ideal location: : :." Line 99 meters: : :. At the surface: : :; at depth: : :. Line 122: on average Line 123: commas before and after 1 cm thick? Line 131: associated with Line 141: replace 'since then' with 'after' Line 215: higher

All the type fixes indicated by reviewer 1 have been taken into account.

Please also note the supplement to this comment:
https://www.clim-past-discuss.net/cp-2018-30/cp-2018-30-AC1-supplement.pdf

[Figure]

**Supplement:**

[revised manuscript text omitted]

SST records from all sites have been previously published (Ruddiman et al.,

1989; Wright and Flower, 2002; Martin-Garcia et al., 2015) and are based in planktonic foraminifers´ census counts.

**3 Results**

*Neogloboquadrina pachyderma* sinistral (Nps) is an indicator of polar water (Cayre et al., 1999; Pflaumann et a., 2003; Eynaud et al., 2009). Except in the eighth climate cycle (MIS19-MIS18), Nps does not vary at glacial-interglacial scale, but peak percentages are associated either to glacial maxima (MIS20) or to deglaciations, both

Terminations and other deglacial events (Fig. 2b), revealing increased advection of polar water at these times. Nps is less abundant during interglacial conditions than during glacials, but it is important to note that glacial Nps´ percentages change through the time series. Nps is more abundant during glacials MIS20, MIS18 (when the highest percentages occurred), and the first half of MIS16, than during late MIS16

and glacial MIS14 (Fig. 2b). After ~650 ka, Nps keeps below 10%, except during some deglacial events, as inferred from sharp decreases in $\delta^{18}$O (Fig.2a-b). This suggests that since mid-MIS16, the polar water only reached the southwest Iberian margin associated to some deglacial episodes, and not during full glacial conditions or glacial maxima, in opposition to what happened before ~650 ka.

*Turborotalita quinqueloba* (Tq) dwells in cold waters and is usually associated with the AF (Johannessen et al., 1994; Cayre et al., 1999). Its percentage in U1385 is lower before MIS16 than since then (Fig. 2c). Highest values occur at ~650 ka and during MIS15b, the glacial interval that interrupted interglacial MIS15. The variation of Tq in site U1385 does not show an interglacial-glacial pattern, which suggests this site did not register the migration of the AF through each climate cycle.

The NAC assemblage (Ottens, 1992; Appendix A) is the most abundant one at this site (Fig. 2), indicating that the ENACWsp dominates the surface oceanography in the area through the time series. This assemblage does not keep a similar interglacial-glacial pattern through the whole study interval, but changes its behaviour at ~650 ka. Previous to ~650 ka, its variation mirrors that of Nps, and the highest values occur during interglacials. In opposition to this, since ~650 ka,, the highest percentages coincide with full glacial conditions (MIS16a and MIS14a), not with interglacials (Fig. 2d).

The Warm Surface (WS) assemblage (Vautravers et al., 2004; Appendix B) is typical of the subtropical water transported eastwards by the AzC. In U1385, this assemblage shows a clear interglacial-glacial pattern only since the seventh climate cycle (Fig. 2e). During glacials previous to MIS16, the WS assemblage is fairly abundant (MIS18), and even more abundant than during interglacials, like in MIS20, when it reaches the highest percentages of the whole study interval. During MIS16, its percentage decreases 
[revised manuscript text omitted]

---

## Author Comment (AC2) · 25 Jul 2018

Referee #2:

The present version of the manuscript is confused and it is very hard to follow the text with these figures. Systematically, the reader has to jump from one figure to another, when it could be possible to plot the data in one single figure.

Figures have been changed as suggested: figures are appropriated called-out, and the information previously included in figures 3 and 4 has been plotted in the new Figure 3

The authors suggest a possible link of the observed changes with change in ciclicity over the Mid-Pleistocene Transition, but a detail discuss on time-series is missing.

[Figure]

The objective of the manuscript is not to study the variation of microfaunal assemblages through a specific time series, but only during glacials before/after the end of the MPT and the completion of the 100-ky cyclicity. Our study focuses on glacials, because the effects of the MPT are more evident during glacial stages, and the surface oceanography in the mid-latitude NAtlantic was similar during interglacials before/after the MPT Anyway, the text has been changed to include time-series description

Moreover, the study interval corresponds to the end of the Mid-Pleistocene Transition and without a detailed spectral and wevelt analysis on proxy records is very hard to propose in the manuscript a connection with this important, but not well understood, climate transition. In my opinion, the authors have to describe, using a statistical approach on proxy data, difference, similitude and trend between the three sites. This statistical approach could be used also to evaluate possible thermal gradients.

We have calculated the thermal gradients that were not included in the original version. Average thermal gradients for each glacial stage have also been calculated, to see if our statements are justified by the data.

The authors plotted as proxies the NAC and WARM SURFACE groups, but the connection with glacial/interglacial cycles is not clear. This is mainly evident for the NAC signal. This signal is characterised by noise and if we exclude, the increase in abundance at ca. 655ka upwards, the signal does not show a particular pattern. The pattern of WARM SURFACE shows a clear strong increase in abundance in correspondence to the onset of interglacial interval. This pattern is not strongly described in the manuscript.

Although our study focuses on glacials, the Results section has been extended to better explain the variations of species and assemblages along the study interval.

The importance of studying the NAC assemblage is the difference between its percentages at site 980 and at site U1385 in figure 3. It is clear that both percentages are similar in interglacials but very different in glacials. This clearly demonstrates the strong influence of the NAC in the high latitudes during interglacials.

There is an explanation for the strong peaks in abundance of N. pachyderma in coincidence of Termination VIII? This peak is in full deglaciation phase.

Yes. In this site, Nps is associated to deglaciations, both Terminations and other main deglacial episodes, as well as to Heinrich-type events (Martin-Garcia et al. 2015). TVIII was very prolonged, with continuous iceberg surges that deposited abundant IRD in the subpolar NAtlantic (e.g., Wright and flower, 2002), and advected very cold water to site U1385, which increased Nps′ percentage.

I would like to suggest to add in the methods a description concerning the construction of the planktonic foraminiferal groups used in the manuscript.

This has been added to the Methods section: "The microfaunal analysis focused on species and assemblages (Appendices A and B) that are associated with North Atlantic surface water masses". The components of each assemblage are included in Appendices, not in Methods because the assemblages are not original of this work, but taken from literature.

I am very surprise to see that Globigerina falconensis is considered as part of warm surface assemblage. This species is generally considered as cool water taxon.

We have used the assemblage defined by Vautravers et al., 2004 (see Appendix B). G. falconensis may be a transitional form, but it has also been identified in tropical waters, as a tropically-adapted symbiont-bearing form of Gb (Hemleben et al., 1989)

In my opinion the strong difference in time resolution of the sites render very difiňĄcult the comparison between the T. quinqueloba and N. pachyderma. In addition, where is the distribution of these taxa for site 607?

The distribution of Nps for site 607 has been added in Fig. 3. It is true that the time resolution between sites does not allow performing certain studies, like detailed statistical analysis, but the existing records allow the comparison with our data and obtain basin-wide conclusions for whole isotope stages.

In addition, the strong difference in NAC patterns from site U1385 and site 980 is not well described and in my opinion not discussed in detail.

The NAC is the dominant assemblage in site U1385 for the whole study interval. On the other hand, site 980 only registers this assemblage when the AF is northward the site. In both sites, the NAC flows from site 607, or its near region. That is the reason why sites 980 and U1385 are compared with site 607, and not between them.

Why Nps is abbreviate? Please write N. pachyderms left coiled – See line 192

As they are continuously mentioned in the text, Neogloboquadrina pachyderma sinistral, as well as Turborotalita quinqueloba, and the assemblages, are abbreviated for sake of making the reading easier.

Line 168 – the authors reported Fig.4c-e, But where is Figure 3?

The appropriate figure has been addressed

Line 213 – Are you sure that the correct figure is 2? I think that the figure to call up is the Fig. 3

The first version of the manuscript did not include the Nps record from site 607, which is why line 213 refers to literature respect to site 607, and to Fig. 2, respect to U1385. Nevertheless, Nps data from site 607 have been plotted in the new Fig. 3 of the reviewed manuscript and the text has been changed accordingly.

Please also note the supplement to this comment:
https://www.clim-past-discuss.net/cp-2018-30/cp-2018-30-AC2-supplement.pdf

[Figure]

**Supplement:**

[revised manuscript text omitted]

SST records from all sites have been previously published (Ruddiman et al.,

1989; Wright and Flower, 2002; Martin-Garcia et al., 2015) and are based in planktonic foraminifers´ census counts.

**3 Results**

*Neogloboquadrina pachyderma* sinistral (Nps) is an indicator of polar water (Cayre et al., 1999; Pflaumann et a., 2003; Eynaud et al., 2009). Except in the eighth climate cycle (MIS19-MIS18), Nps does not vary at glacial-interglacial scale, but peak percentages are associated either to glacial maxima (MIS20) or to deglaciations, both

Terminations and other deglacial events (Fig. 2b), revealing increased advection of polar water at these times. Nps is less abundant during interglacial conditions than during glacials, but it is important to note that glacial Nps´ percentages change through the time series. Nps is more abundant during glacials MIS20, MIS18 (when the highest percentages occurred), and the first half of MIS16, than during late MIS16

and glacial MIS14 (Fig. 2b). After ~650 ka, Nps keeps below 10%, except during some deglacial events, as inferred from sharp decreases in $\delta^{18}$O (Fig.2a-b). This suggests that since mid-MIS16, the polar water only reached the southwest Iberian margin associated to some deglacial episodes, and not during full glacial conditions or glacial maxima, in opposition to what happened before ~650 ka.

*Turborotalita quinqueloba* (Tq) dwells in cold waters and is usually associated with the AF (Johannessen et al., 1994; Cayre et al., 1999). Its percentage in U1385 is lower before MIS16 than since then (Fig. 2c). Highest values occur at ~650 ka and during MIS15b, the glacial interval that interrupted interglacial MIS15. The variation of Tq in site U1385 does not show an interglacial-glacial pattern, which suggests this site did not register the migration of the AF through each climate cycle.

The NAC assemblage (Ottens, 1992; Appendix A) is the most abundant one at this site (Fig. 2), indicating that the ENACWsp dominates the surface oceanography in the area through the time series. This assemblage does not keep a similar interglacial-glacial pattern through the whole study interval, but changes its behaviour at ~650 ka. Previous to ~650 ka, its variation mirrors that of Nps, and the highest values occur during interglacials. In opposition to this, since ~650 ka,, the highest percentages coincide with full glacial conditions (MIS16a and MIS14a), not with interglacials (Fig. 2d).

The Warm Surface (WS) assemblage (Vautravers et al., 2004; Appendix B) is typical of the subtropical water transported eastwards by the AzC. In U1385, this assemblage shows a clear interglacial-glacial pattern only since the seventh climate cycle (Fig. 2e). During glacials previous to MIS16, the WS assemblage is fairly abundant (MIS18), and even more abundant than during interglacials, like in MIS20, when it reaches the highest percentages of the whole study interval. During MIS16, its percentage decreases 
[revised manuscript text omitted]

---

## Author Response (AR2)

**Cp-2018-30_2ⁿᵈ. Revision COVER LETTER**

Gloria M. Martin-Garcia
Departamento de Geología
Universidad de Salamanca
Plaza de los Caídos s/n
37008 Salamanca, Spain
gm.martin@usal.es

September 2018

To Marit-Solveig Seidenkrantz, Editor,

Re: 2ⁿᵈ revision of the manuscript **cp-2018-30**
    (by Gloria M. Martin-Garcia, Francisco J. Sierro, José A. Flores and Fatima Abrantes)

    Dear Editor, we are submitting the new version of our manuscript, that has been reviewed according to the Referees´ suggestions.

    A point-to-point reply to Referees´ comments is detailed in the "*2ⁿᵈ Response to Ref*" files. The main changes in this version of the manuscript are:

The title of the manuscript has been changed into: "***Change in the North Atlantic circulation associated to the mid-Pleistocene transition***", to avoid misleading the reader, as Ref #1 suggests.

The abbreviation of species have been changed as Ref # 2 suggests

Line 47:
    The following cites have been added: (McCartney and Talley, 1984; Ruddiman and McIntyre, 1984; Schmitz and McCartney, 1993; Rahmstorf, 1994; Chapman and Maslin, 1999)

Lines 68/69, 70, 89, 100, 102, 150, 156, 157, 177, 283, 313:
    The text of these lines has been modified as suggested

Line 71/72:
    The text has been changed as follows: "the Earth´s climate system underwent a major change, non-linear 100 ky cycles appeared and superimposed over the more linear, orbital ones of 41 and 23 ky."

Lines 121-124:
    The age models for sites 980 and 607 have been explained (new lines 141-143)

Line 131:
    A detailed list of indicator species has been included in this section

Line135-141:
    The text has been changed as follows (new Lines 175-178: "This estimation of thermal gradients is possible because all the SST records used for this work are based in planktonic foraminifers´ census counts. Nevertheless, previous to the comparison, interpolation was applied to obtain records with the same age points."

Line 142:
    The wrong paragraph has been deleted

Line 155:
    The paragraph has been changed as follows: "Sediments at Site U1385 define a single, very uniform, lithological unit. Calcareous muds and calcareous clays dominate the lithology. The relative proportions of carbonate (23% - 39%) and terrigenous materials show in the sediment color that varies from dark (i.e., more terrigenous) to light (i.e., more calcareous). The average sedimentation rate for the section is of ~10 cmky$^{-1}$ (Stow et al., 2012)."

Lines 176-180:

The new paragraph states as follows: "In U1385, this assemblage shows a clear interglacial-glacial pattern only since Termination TVIII, its percentage decreasing gradually during MIS17-16 until the glacial maximum (Fig. 2e). Comparing glacial stages, MIS20 records the highest average relative abundance (16.8%) and MIS14, the lowest (8.7%). Termination TIX records the most abrupt decrease of this assemblage (15% drop), while at TVI it even increases (5% rise). At the beginning of each interglacial, the percentage of this assemblage rises rapidly, suggesting that the AzC strengthens rapidly in the area after Terminations."

Line 314:

The former reference was wrong; the right one is Martin-Garcia et al. 2015, and has been included in the new version

Looking forward hearing from you soon,
Yours sincerely

Gloria M. Martin-Garcia (on behalf of all the authors of the paper)

**RESPONSE TO REVIWERS**

**Response to Referee # 1 // report #2**

The manuscript by Martin-Garcia et al. is improved, but I believe that it needs additional revisions. The authors appear to have addressed the majority of my comments, but not all. It is a bit difficult to determine because the cover letter does not contain references to the specific line numbers where the changes have been implemented so that one can easily cross-check. In addition, the manuscript text is still unclear at times due to the awkward use of the English language.

It occurred to me that the title is a bit misleading. If a study examines the role of North Atlantic circulation in the mid-Pleistocene transition, then I would expect that it spans the entire interval of time. Here the records do not begin until after the mid-point (MIS 20). So only half of the event is captured.

The title of the article has been changed into "Change in the North Atlantic circulation associated to the mid-Pleistocene transition" to avoid misleading the reader.

I still think that a mention of the closed sum problem is important. In my mind changes in a dominant assemblage can drive the percentages of a less dominant species. True, this approach is often used. But that does not mean that it does not have uncertainties.

We know that his problem is intrinsic to all ecological and paleoecological studies. A decrease in abundance of individuals of one species will increase the relative abundance of the others, but there is no other way to deal with this. Absolute abundances of the planktonic species cannot be used in paleoecology, because they are determined by other, non-ecological parameters, such as dilution by detrital inputs, etc.

Regarding my comment to Line 251 of the original manuscript (now section 5.3): The authors maintain that there is an increase in NADW formation rate during glacial intervals begin with MIS 22 and ending with MIS 14. They cite Wright and Flower, 2002; Hodell et al., 2008,

Poirier and Billups 2014, Hodell et al., 2015 for this statement (in the text or rebuttal). I have looked at all of these articles and cannot find a single statement to this effect. In fact, Wright and Flower cite Raymo et al., 1997 saying that there is greater suppression of NADW …. from 950 to 350 Ka. Hodell et al 2015 is about the age model for Site U1385, and I don't see a discussion of deep water circulation in that paper, the d13C record is not shown. Hodell and Channel (2016) note that d13C minima increase from MIS 22 to MIS 14, and they cite Raymo for this observation. However, to my reading of this article, they do not say that this is due to an increase in NADW formation.

This interpretation, as we say at the beginning of the section, is based in "a close correlation between the rate of AMOC and benthic $d^{13}C$ levels (Zahn et al, 1997; Adkins et al., 2005; Hoogakker et al., 2006)". From this assumption, we interpret the published d13C data.

D13C, compared with d18O, from sites 980 (Fig 3 in W&F 2002), U1308-607 (fig 10 in Hodell et al. 2008) (Fig 3 in Hodell-Channell 2016), 1063 (Fig 4, 5c, 7d-h in P&B 2014) and U1385 (Fig 4 in Martin-Garcia et al., 2015), document this increase. We do not think there is any doubt respect MIS 14. As for MIS 16, although it was a more prolonged and severe (in ice volume) glacial than MIS 18, 20 and 22, its d13C was not so low as should be expected. Besides, during MIS 16 glacial maximum, d13C was higher than during the previous glacial maxima.

Respect to Hodell et al 2015, the referee is right. D13C from site U1385 appeared in Martin-Garcia et al., 2015, instead in the previously cited paper. The newly reviewed manuscript has the correct reference.

The rest of my comments are in order of occurrence in the manuscript text:
Line 47: add a citation?

The following cites have been added: (McCartney and Talley, 1984; Ruddiman and McIntyre, 1984; Schmitz and McCartney, 1993; Rahmstorf, 1994; Chapman and Maslin, 1999)

Line 68/69: related to the
Line 70: no comma after which

These lines have been modified as suggested

Line 71/72: the climate system did not switch, the 100 kyr cycle started to appear. The 41 and 23 kyr cycles continue to be present

The text has been changed as follows: "the Earth´s climate system underwent a major change, non-linear 100 ky cycles appeared and superimposed over the more linear, orbital ones of 41 and 23 ky."

Line 80: why glacial? The manuscript seems to address both, glacial and interglacial intervals

Enhanced ice sheets growth and reduced NADW formation are postulated as partly responsible for the change of the climate system phasing (Imbrie et al., 1993). This manuscript explores such possibility by studying variations in the advection of warm and saline water to subpolar latitudes, before/after the occurrence of the first 100 ky cycle. We are not so interested in the potential change of circulation during interglacials before/after because they did not contribute to increase the ice volume.

The comparison with interglacial conditions serves to highlight the change of circulation that occurred during glacials.

Line 89: I would replace the word 'proven' with something like 'shown'
Line 100: of the Quaternary
Line 102 at the surface

In these lines, the text has been changed as suggested

Line 155: No single sentence paragraphs. What is the difference between a calcareous mud and a calcareous clay? Calcareous means biogenic carbonate as well, right? This paragraph needs a bit rewording to make it more clear.

The difference between mud and clay is the grain size.

The paragraph has been changed as follows:

"Sediments at Site U1385 define a single, very uniform, lithological unit.  Calcareous muds and calcareous clays dominate the lithology. The relative proportions of carbonate (23% - 39%) and terrigenous materials show in the sediment color that varies from dark (i.e., more terrigenous) to light (i.e., more calcareous). The average sedimentation rate for the section is of ~10 cmky$^{-1}$ (Stow et al., 2012)."

Lines 121-124: What about the age models of the other sites? For calculation of the thermal gradient it is really important that the age models are comparable.

New age models were calculated for sites 980 and 607, based on correlations with the LR04 stack. (Lines 141-143)

Line 131: More detail? I think that it would be appropriate to include a brief list of warm versus cold assemblages etc., then refer to the Appendix with more details

A detailed list of indicator species has been included in this section

Line135-141: How are the SSTs subtracted? Are the records interpolated to the same ages? The method needs more detail.

The method has been explained in more detail.

Lines 175-178: "This estimation of thermal gradients is possible because all the SST records used for this work are based in planktonic foraminifers´ census counts. Nevertheless, previous to the comparison, interpolation was applied to obtain records with the same age points."

Line 142: SSTS from Site 607 and 980 have been published. Not all SSTs records have been published as you are presenting those from Site 1385 here for the first time?

The reviewer is right, and the wrong paragraph has been deleted

Line 150: replace the two 'to' with 'with'
Line 156: replace 'keeps' with 'stays'
Line 157: specify which ones, 'some' is vague

In these lines, the text has been changed as suggested

Lines 167-174: To my eye the variations are subtle. It seems very qualitative

As the NAC assemblage is the dominant one during the whole interval, and its percentage is never lower than 30 %, subtle variations can indicate substantial changes of the surface circulation

Line 177: instead of referring to a particular climate cycle, just say which MIS, or which Termination? Those are labeled in the figure and it is thus easier to find.

The text has been modified as suggested: "since Termination TVIII"

Line 177: 'since' is not correct as the records end at MIS 14.

As we are detailing our results, and the study interval has already been defined, it is understood that "since" refers to our record.

Lines 176-180: I am not sure I can follow the sentence at all. MIS 20 is not an interglacial. And, the variations are subtle, what do you mean with' fairly'? I would suggest providing the changes of the percentages to give the reader a sense of how much of a change is actually occurring.

The text has been re-written to clarify it.

It states as follows: "In U1385, this assemblage shows a clear interglacial-glacial pattern only since Termination TVIII, its percentage decreasing gradually during MIS17-16 until the glacial maximum (Fig. 2e). Comparing glacial stages, MIS20 records the highest average relative abundance (16.8%) and MIS14, the lowest (8.7%). Termination TIX records the most abrupt decrease of this assemblage (15% drop), while at TVI it even increases (5% rise). At the beginning of each interglacial, the percentage of this assemblage rises rapidly, suggesting that the AzC strengthens rapidly in the area after Terminations."

Line 244: is a 0.2 per mil difference in d13C values really significant in terms of NADW? There are other factors that determine the d13C values of benthic forams.

Maximal variation of d13C during MIS 15-14, for instance, was 1 per mil. Although it is true that other factors can contribute, we did not find anything that definitely demonstrates that variations in d13C were NOT related with the presence of different masses of water at the bottom.

Line 283: associated with

The text has been modified as suggested

Lines 285-292: see comment above. I cannot find any statements to this effect in the literature cited in this section. The authors seem to base this on a slight increase in benthic foram d13C values at their site? I think the interpretation of the d13C record is a lot more complex as presented in this study. Do these studies really say that there is an increase in NADW during glacial intervals specifically? Or are they referring to a more general increase in NADW over time, which includes interglacial intervals? As noted above, I cannot find d13C records in the Hodell et al 2015 paper, do you mean Hodell and Channel 2016? They describe an increase in d13C values, but I don't think they discuss deep water circulation.

This question was answered above.

Our statements are not only based on the record from U1385, but also on the data shown in the cited articles (although the authors not necessarily discuss deep water circulation)

As for the increase in NADW formation during interglacials, as well as glacials, we are not arguing against it. The issue we address is that the enhanced AMOC during glacials MIS 16 and MIS 14, had consequences in the building of ice sheets, and prolonged the duration of the glacial stages

The cite Hodell et al 2015 was wrong; d13C data from site U1385 appeared in Martin-Garcia et al 2015. The right reference has been added.

Line 313: southern-more or more southern?

The text has been modified as suggested

Line 314: I cannot find any reference to d13C in the Hodell et al. 2015 paper.

The right reference is Martin-Garcia et al. 2015 (Figure 4)

Line 315: How can you tell from your data that the overturning cell was deepening? Hodell and Channel describe that d13C minima get higher between MIS 22 and MIS14 citing Raymo. But I don't think that they say that this is due to more NADW formation. In any case, there could be other factors.

As explained above, this interpretation is based in the correlation between d13C and masses of water. Although it is true that other factors can contribute, we did not find anything that definitely demonstrates that variations in d13C were NOT related with the presence of different masses of water at the bottom.

Assuming this, d13C records from different NAtlantic sites, not only from U1385, show a progressively increased presence of NADW at depths previously occupied by the AABW (Fig 3 in Hodell-Channell 2016; Fig 4, 5c, 7d-h in P&B 2014; Fig 4 in Martin-Garcia et al., 2015)

**Response to Referee # 2 // report #2**

I have just two technical comments, as follows:

i) I do not like to use only letters for planktonic foraminifera (i.e., Turborotalita quinqueloba= Tq). Because of in the manuscript there are several abbreviation related to the different water fronts, the authors can use the classic abbreviation for planktonic forams, ie., T. quinqueloba. The same for Neogloboquadrina pachyderma left coiled.

The abbreviations of species have been changed as suggested ii) the authors at chapter 5.1 start for fig. 3f. In my opinion, the first cited figure in a chapter has to be fig. 3a and after fig. 3b, fig. 3c…….

Yes, this order is the most common one. Nevertheless, panels in figure 3 are arranged in what we think is a more logical and intuitive way, as follows (from bottom to top):

1. Data from the bottom (d18O and d13C)

2. Data of species used as indicators (from polar to mid-latitude)

3. SST

4. Thermal gradient obtained from the previous

[revised manuscript text omitted]